# Fast Node Embeddings: Learning Ego-Centric Representations

## Abstract

Representation learning is one of the foundations of Deep Learning and allowed important improvements on several Machine Learning tasks, such as Neural Machine Translation, Question Answering and Speech Recognition. Recent works have proposed new methods for learning representations for nodes and edges in graphs. Several of these methods are based on the SkipGram algorithm, and they usually process a large number of multi-hop neighbors in order to produce the context from which node representations are learned. In this paper, we propose an effective and also efficient method for generating node embeddings in graphs that employs a restricted number of permutations over the immediate neighborhood of a node as context to generate its representation, thus ego-centric representations. We present a thorough evaluation showing that our method outperforms state-of-the-art methods in six different datasets related to the problems of link prediction and node classification, being one to three orders of magnitude faster than baselines when generating node embeddings for very large graphs.

## 1 Introduction

Many important problems involving graphs require the use of learning algorithms to make predictions about nodes and edges. These predictions and inferences on nodes and edges from a graph are typically done using classifiers with carefully engineered features (Grover & Leskovec, 2016). These features, besides taking time and manual labor to be developed and acquired, usually do not generalize well to other problems or contexts.

The field of Natural Language Processing (NLP) has had many advances due to the use of algorithms that learn word representations, instead of manually extracted features. Originally proposed by Bengio et al. (2003) and commonly used with Word2Vec algorithms like CBOW and SkipGram (Mikolov et al., 2013a), word embeddings are used in many state-of-the-art solutions for neural machine translation (Luong & Manning, 2016; Firat et al., 2016), question answering (Andreas et al., 2016) and natural language generation (Wen et al., 2015). Recent works have proposed new methods for learning representations for nodes and edges in graphs, based on random walks (Perozzi et al., 2014; Grover & Leskovec, 2016) or auto-encoding adjacency vectors (Wang et al., 2016).

In this paper, we propose a new general purpose method for generating node embeddings in very large graphs, which we call Neighborhood Based Node Embeddings (or simply NBNE). NBNE is based on the SkipGram algorithm and uses nodes neighborhoods as contexts. NBNE outperforms state-of-the-art DeepWalk (Perozzi et al., 2014) and Node2Vec (Grover & Leskovec, 2016) for the tasks of link prediction and node classification on six collections, being one to three orders of magnitude faster. In this work, we considered DeepWalk and Node2Vec as baselines.

The main reason for this improvement on effectiveness and efficiency is that we concentrate learning on the "predictable" parts of the graph. A study by Facebook research (Edunov et al., 2016) found that each person in the world (at least among the people active on Facebook) is connected to every other person by an average 3.57 other people. In a graph of this magnitude and connectedness, learning node embeddings by maximizing the log-probability of predicting nearby nodes in a random walk (with a window size of 5) can be highly inefficient and make it 'harder' for the embeddings to be constructed, even if these random walks are biased like in Node2Vec. We suspect this can also make them more unstable, which would explain why they need more iterations before embedding convergence.

The main contributions of this work are:

- We present a **new general purpose method** that is more effective and more efficient than state-of-the-art methods for generating node embeddings in graphs.

- Experimental results in solving the link prediction and node classification problems for six graph sets show that **our method outperforms, in terms of effectiveness and efficiency**, the methods DeepWalk and Node2Vec.

- In spite of the fact that our method has the same time complexity of the two baselines DeepWalk and Node2Vec, we were able to **improve the training time by one to three orders of magnitude**, which is important when dealing with very large graphs. For instance, to learn node embeddings for a graph containing 317,080 nodes and 1,049,866 edges, collected from the DBLP[1] repository (Yang & Leskovec, 2012), NBNE took approximately 14m30s minutes, DeepWalk approximately 164m34s and Node2Vec approximately 3,285m59s (more than 54 hours).

- We provide a thorough evaluation of our method in real and synthetic graphs, **motivating our choice for a semi supervised algorithm**. Our method has a single tunable parameter (number of permutations, which will be explained later) that can be tuned at once on the training set to avoid overfitting the representations.

## 2  RELATED WORK

The definition of node similarity and finding general purpose node and/or edge representations are non-trivial challenges (Lü & Zhou, 2011). Many definitions of similarity in graphs use the notion of first and second order proximity. First-order proximity is the concept that connected nodes in a graph should have similar properties, while the second-order proximity indicates that nodes with similar neighborhoods should have common characteristics.

Some earlier works on finding these embeddings use various matrix representations of the graph, together with dimensionality reduction techniques, to obtain the nodes' representations (Roweis & Saul, 2000; Tenenbaum et al., 2000). A problem with these approaches is that they usually depend on obtaining the matrix' eigenvectors, which is infeasible for large graphs ($O(n^{2.376})$) with the Coppersmith-Winograd algorithm (Coppersmith & Winograd, 1987)). Recent techniques attempt to solve this problem by dynamically learning representations for nodes in a graph using non-linear techniques based either on first and second order proximities (Tang et al., 2015; Wang et al., 2016) or random walks (Perozzi et al., 2014; Grover & Leskovec, 2016).

Other recent works focus on finding representations for specific types of graphs. TriDNR (Pan et al., 2016) uses a graph structure together with node content and labels to learn node representations in two citation networks. Their work can be directly applied to any graph where nodes have labels and/or text contents. TEKE (Wang & Li, 2016) and KR-EAR (Lin et al., 2016) find representations for entities in knowledge graphs and metapath2vec (Dong et al., 2017) finds node representations in heterogeneous networks. The method LINE (Tang et al., 2015) finds a $d$ dimensional representation for each node based on first and second-order graph proximities, not being feasible for large graphs, because its cost function depends on the whole adjacency matrix ($O(|V|^2)$).

Another method, Structural Deep Network Embedding (SDNE) (Wang et al., 2016), is also based on first and second order proximities. It uses autoencoders to learn a compact representation for nodes based on their adjacency matrix (second-order proximity), while forcing representations of connected nodes to be similar (first-order proximity) by using an hybrid cost function. SDNE is also not feasible for large graphs, since the autoenconders are trained on the complete adjacency vectors. Each vector has size $O(|V|)$ and is created at least once, creating a lower bound on time complexity $O(|V|^2)$.

The method DeepWalk (Perozzi et al., 2014) generates $k$ random walks starting on each vertex in the graph to create sentences where each "word" is a node. These sentences are then trained using the SkipGram algorithm to generate node embeddings. This method has a time complexity bounded by $O(|V| \log |V|)$.

---

[1]http://dblp.uni-trier.de

Node2Vec (Grover & Leskovec, 2016) also uses random walks with SkipGram and can be seen as a generalization of DeepWalk. The difference between the two methods is that Node2Vec's walks are random, but biased by two pre-assigned parameters $p$ and $q$. During the creation of the walks, these parameters are used to increase the chance of the walk returning to a parent node or going farther from it. This method uses a semi-supervised approach which requires several models to be generated and a small sample of labeled nodes to be used so that the best parameters $p$ and $q$ can be chosen. Node2Vec is not efficient for densely connected graphs, since its time and memory dependencies on the graph's branching factor $b$ are $O(b^2)$.

In this work, we considered DeepWalk (Perozzi et al., 2014) and Node2Vec (Grover & Leskovec, 2016) as baselines, since they are scalable, having a time complexity ($O(|V|\log|V|)$). The main differences between NBNE and the two baselines are: (i) we use a different sentence sampling strategy which is based in a node's neighborhood instead of random walks, (ii) NBNE is more effective than both Node2Vec and DeepWalk, as supported by our experiments in six different datasets, and (iii) NBNE is efficient for both dense and sparse graphs and scalable for very large applications, having a faster training time than both Node2Vec and DeepWalk.

## 3 NEIGHBORHOOD BASED NODE EMBEDDINGS

The context of a word is not a straightforward concept, but it is usually approximated by the words surrounding it. In graphs, a node's context is an even more complex concept. As explained above, DeepWalk and Node2Vec use random walks as sentences and consequently as contexts in which nodes appear.

In this work, the contexts are based solely on the neighborhoods of nodes, defined here as the nodes directly connected to it, focusing mainly on the second-order proximities. Consequently, nodes' representations will be mainly defined by their neighborhoods and nodes with similar neighborhoods (contexts) will be associated with similar representations.

### 3.1 SENTENCE GENERATION

In our Neighborhood Based Node Embedding's (NBNE) method, as the name implies, sentences are created based on the neighborhoods of nodes. There are two main challenges in forming sentences from neighborhoods, as follows:

- A sentence containing all the neighbors from a specific highly connected root node might be of little use. Most neighbors would be distant from each other in the sentence, not influencing each other's representations, and not directly influencing the root node.

- There is no explicit order in the nodes in a neighborhood. So there is no clear way to choose the order in which they would appear in a sentence.

In this work, the solution is to form small sentences, with only $k$ neighbors in each, using random permutations of these neighborhoods. Algorithm 1 presents the code for generating sentences. As a trade-off between training time and increasing the training dataset the user can select the number of permutations $n$. Selecting a higher value for $n$ creates a more uniform distribution on possible neighborhood sentences, but also increases training time.

---

**Algorithm 1** Sentence Sampling

---

1: **procedure** GETSENTENCES(graph, $n$)
2:     $sentences \leftarrow [\emptyset]$
3:     **for** $j$ in $0 : n$ **do**
4:         **for** $node$ in $graph$.nodes() **do**
5:             $neighbors \leftarrow$ random_permutation($node$.neighbors())
6:             **for** $i$ in $0 : \text{len}(neighbors)/k$ **do**
7:                 $sentence \leftarrow [node] + neighbors[i \cdot k : i \cdot (k+1)]$
8:                 $sentences$.append($sentence$)

---

## 3.2 Learning Representations

As described in Section 3.1, Algorithm 1 forms a set of sentences $S$, where each word is actually a node from the graph. We train the vector representations of nodes by maximizing the log probability of predicting a node given another node in a sentence and given a set of representations $r$. We use a window of size $k$ which is equal to the size of the generated sentences, so that each node in a sentence predicts all the others. The log probability maximized by NBNE is given by:

$$\max_r \quad \frac{1}{|S|} \sum_{s \in S} \left( \log \left( p \left( s | r \right) \right) \right) \tag{1}$$

where $p \left( s | r \right)$ is the probability of each sentence, which is given by:

$$\log \left( p \left( s | r \right) \right) = \frac{1}{|s|} \sum_{i \in s} \left( \sum_{j \in s, j \neq i} \left( \log \left( p \left( v_j | v_i, r \right) \right) \right) \right) \tag{2}$$

where $v_i$ is a vertex in the graph and $v_j$ are the other vertices in the same sentence. The probabilities in this model are learned using the feature vectors $r_{v_i}$, which are then used as the vertex representations. The probability $p \left( v_j | v_i, r \right)$ is given by:

$$p \left( v_o | v_i, r \right) = \frac{\exp \left( r\prime_{v_o}^T \times r_{v_i} \right)}{\sum_{v \in V} \left( \exp \left( r\prime_v^T \times r_{v_i} \right) \right)} \tag{3}$$

where $r\prime_{v_j}^T$ is the transposed output feature vector of vertex $j$, used to make predictions. The representations $r\prime_v$ and $r_v$ are learned simultaneously by optimizing Equation (1). This is done using stochastic gradient ascent with negative sampling (Mikolov et al., 2013b).

By optimizing this log probability, the algorithm maximizes the predictability of a neighbor given a node, creating node embeddings where nodes with similar neighborhoods have similar representations. Since there is more than one neighbor in each sentence, this model also makes connected nodes have similar representations, because they will both predict each others neighbors, resulting in representations also having some first order similarities. A trade-off between first and second order proximity can be achieved by changing the parameter $k$, which simultaneously controls both the size of sentences generated and the size of the window used in the SkipGram algorithm. A further discussion on this effect can be seen in Appendix B.3.

## 3.3 Avoiding Overfitting Representations

When using large values of $n$ (i.e., number of permutations) on graphs with few edges per node, some overfitting can be seen on the representations, as shown in details in Section 5.1 and in Appendix B.2. This overfitting can be avoided by sequentially training on increasing values of $n$ and testing the embeddings on a validation set every few iterations, stopping when performance stops improving, as shown in Algorithm 2.

---

**Algorithm 2** NBNE without Overfitting

---

1: **procedure** TRAINNBNE(graph, max_n)
2:     $sentences \leftarrow$ get_sentences($graph, max\_n$)
3:     $model \leftarrow$ [initialize_model()]
4:     **for** $j$ in $0 : log_2(max\_n)$ **do**
5:         $model \leftarrow$ train($model, sentences[2^j : 2^{j+1}]$)
6:         $error \leftarrow$ test($new\_model, validation\_set$)
7:         **if** $error > old\_error$ **then**
8:             break

---

## 4 Experiments

NBNE was evaluated on two different tasks: link prediction, and node classification.[2] We used a total of six graph datasets to evaluate NBNE and Table 1 presents details about these datasets. A

---

[2]Link to NBNE code will be made available in final publication.

Table 1: Statistics on the first six graph datasets

|  | Nodes | Edges | Edges/Node | # Classes |
|---|---|---|---|---|
| Facebook[1] (McAuley & Leskovec, 2012) | 4,039 | 88,234 | 21.84 | - |
| Astro[1] (Leskovec et al., 2007) | 18,772 | 198,110 | 10.55 | - |
| PPI[1,2] (Breitkreutz et al., 2008) | 3,890 | 38,739 | 9.95 | 49 |
| Wikipedia[1,2] (Mahoney, 2011) | 4,777 | 92,517 | 19.36 | 39 |
| Blog[1,2] (Zafarani & Liu, 2009) | 10,312 | 333,983 | 32.38 | 39 |
| DBLP[1] (Yang & Leskovec, 2012) | 317,080 | 1,049,866 | 3.31 | - |

[1] used in Link Prediction
[2] used in Node Classification

brief description of each dataset can be found in Appendix A and an analysis of their assortativity properties in Appendix B.1. We present results for the link prediction problem in Section 4.1 and for the node classification problem in Section 4.2. For all experiments we used sentences of size $k = 5$ and embeddings of size $d = 128$, while the number of permutations was run for $n \in \{1, 5, 10\}$. The best value of $n$ was chosen according to the precision on the validation set and we used early stopping, as described in Section 3.3.

On both these tasks, DeepWalk and Node2Vec were used as baselines, having been trained and tested under the same conditions as NBNE and using the parameters as proposed in (Grover & Leskovec, 2016). More specifically, we trained them with the same training, validation and test sets as NBNE and used a window size of 10 ($k$), walk length ($l$) of 80 and 10 runs per node ($r$). For Node2Vec, which is a semi-supervised algorithm, we tuned $p$ and $q$ on the validation set, doing a grid search on values $p, q \in \{0.25; 0.5; 1; 2; 4\}$. We also evaluated NBNE on two synthetic graphs with different sizes and sparseness, which can be seen on Appendix C, and an author name disambiguation task, on Appendix F. A comparison between NBNE and SDNE can be seen on Appendix D.

## 4.1 LINK PREDICTION

**Setup.** Link prediction attempts to estimate the likelihood of the existence of a link between two nodes, based on observed links and the nodes' attributes (Lü & Zhou, 2011). Typical approaches to this task use similarity metrics, such as Common Neighbors or Adamic-Adar (Adamic & Adar, 2003). Instead of these hand made similarity metrics, we propose to train a logistic classifier based on the concatenation of the embeddings from both nodes that possibly form an edge and predict the existence or not of the edge.

To train NBNE on this task, we first obtained a sub-graph with 90% randomly select edges from each dataset, and obtained the node embeddings by training NBNE on this sub-graph. We, then, separated a small part of these sub-graph edges as a validation set, using the rest to train a logistic regression with the learned embeddings as features.

After the training was completed, the unused 10% of the edges were used as a test set to predict new links. 10-fold cross-validation was used on the entire training process to access the statistical significance of the results, analyzing statistical difference between the baselines and NBNE. To evaluate the results on this task, we used as metrics: AUC (area under the ROC curve) (Baeza-Yates & Ribeiro-Neto, 2011), and training time.[3] The logistic regressions were all trained and tested using all available edges (respectively in the training or test set), and an equal sized sample of negative samples, which, during training, included part of the 10% removed edges.

**Results.** Table 2 presents results for this task. Considering AUC scores on the Link Prediction task, NBNE was statistically better[4] than both DeepWalk and Node2Vec on the Astro and PPI datasets, with more than 7% improvement, also showing a 4.67% performance gain in Wikipedia and a small, but statistically significant, gain on Blog. Only losing by a small percentage on Facebook, with a difference that was not statistically significant.

---

[3]Training times were all obtained using 16 core processors, running NBNE, Node2Vec or DeepWalk on 12 threads, with all algorithms having been implemented using gensim (Řehůřek & Sojka, 2010). More detailed results including precision on the training and test sets can be seen in Appendix E.

[4]In all experiments we performed Welch's t-tests with $p = 0.01$. The symbol $^*$ marks results which are statistically different from NBNE.

Table 2: Link prediction results

| | Facebook | | Astro | | PPI | |
|---|---|---|---|---|---|---|
| | AUC | Training Time | AUC | Training Time | AUC | Training Time |
| NBNE | 0.9688 | **0m11s** | **0.8328** | **0m07s** | **0.8462** | **0m02s** |
| DeepWalk | 0.9730 | 2m26s | 0.7548* | 6m55s | 0.7741* | 2m30s |
| Node2vec | **0.9762** | 69m33s | 0.7738* | 182m16s | 0.7841* | 66m37s |
| Gain | -0.76% | 12.96x 369.85x | 7.62% | 59.06x 1555.80x | 7.91% | 77.43x 2061.67x |

| | Wikipedia | | Blog | | DBLP | |
|---|---|---|---|---|---|---|
| | AUC | Training Time | AUC | Training Time | AUC | Training Time |
| NBNE | **0.6853** | **0m02s** | **0.9375** | **1m11s** | **0.9335$^\dagger$** | **14m30s** |
| DeepWalk | 0.6534* | 7m38s | 0.9098* | 28m13s | 0.9242$^\ddagger$ | 164m34s |
| Node2Vec | 0.6547* | 236m60s | 0.9202* | 838m41s | 0.9322$^\ddagger$ | 3,285m59s |
| Gain | 4.67% | 194.86x 6049.77x | 1.88% | 23.86x 709.24x | 0.13% | 11.34x 226.52x |

$\dagger$ average of 10 fold results
$\ddagger$ no statistical tests were run, due to the time necessary to run a single fold

In DBLP, NBNE again presents the best AUC score, although this difference was small and its statistical significance could not be verified due to the large training times of the baselines. This dataset contains the largest graph analyzed in this work (317,080 nodes and 1,049,866 edges) and in it, to train a single fold, Node2Vec took 3,285m59s (more than 54 hours) and DeepWalk took 164m34s (approximately 2 hours and 44 minutes), while NBNE took only 14m30s, which represents a 226/11 times improvement over Node2Vec and DeepWalk, respectively.

Considering training time for this task, NBNE presents the biggest improvements on sparser networks of medium size, like Astro, PPI and Wikipedia datasets. On these graphs, the best results are for $n = 1$, resulting in more than 50x faster training than DeepWalk and more than 1,500 times faster than Node2Vec, achieving a 6,049 times faster training than Node2Vec on Wikipedia. For the Blog and Facebook datasets the best results are for $n = 5$, resulting in larger training times, but still more than one order of magnitude faster than DeepWalk and more than 350 times faster than Node2Vec. For the DBLP dataset, the best results were achieved with $n = 10$, still much faster than the baselines.

## 4.2 NODE CLASSIFICATION

**Setup.** Given a partially labeled graph, node classification is the task of inferring the classification of the unknown nodes, using the structure of the graph and/or the properties of the nodes. In this task, the node embeddings were trained using NBNE on the complete graph. After obtaining the node embeddings, 80% of the labeled nodes in the graph were used to train a logistic classifier that predicted the class of each node, while 5% of the nodes were used for validation and the remaining 15% nodes were used as a test set. This test was repeated for 10 different random seed initializations to access the statistical relevance of the results.

**Results.** Results on the Blog, PPI and Wikipedia datasets are shown in Table 3 and are presented in terms of Macro F1 scores and training times. NBNE produces statistically similar results to its baselines, in terms of Macro F1, on both PPI and Wikipedia, while showing a statistically significant 22.45% gain in the Blog dataset, indicating that NBNE's embeddings did not only get a better accuracy on Blog, but also that correct answers were better distributed across the 39 possible classes.

Considering training times, NBNE is more than 10 times faster than DeepWalk on these three datasets and is $[300 \sim 600]$ times faster than Node2Vec. NBNE didn't show statistically worse result in any dataset analyzed here[5], while having an order of magnitude faster training time than DeepWalk and more than two orders of magnitude faster training time than Node2Vec.

---

[5]Except for test precision on Wikipedia, losing to DeepWalk. For more details, see Appendix E

Table 3: Node classification results

|  | Blog | | PPI | | Wikipedia | |
|---|---|---|---|---|---|---|
|  | Macro F1 | Training Time | Macro F1 | Training Time | Macro F1 | Training Time |
| NBNE | **0.2004** | **1m57s** | 0.0978 | **0m16s** | **0.0727** | **0m41s** |
| DeepWalk | 0.1451* | 31m31s | **0.0991** | 3m04s | 0.0679 | 13m04s |
| Node2vec | 0.1637* | 959m12s | 0.0971 | 83m02s | 0.0689 | 408m00s |
| Gain | 22.45% | 16.18x 492.57x | -1.35% | 11.82x 319.78x | 5.56% | 19.04x 594.62x |

## 5 FURTHER ANALYSIS

### 5.1 NUMBER OF PERMUTATIONS ($n$)

The quality of NBNE's embeddings depends on both the size of the embeddings ($d$) and the number of permutations ($n$). For highly connected graphs, larger numbers of permutations should be chosen ($n \in [10, 1000]$) to better represent distributions, while for sparser graphs, smaller values can be used ($n \in [1, 10]$).

Figure 1 shows AUC scores versus embedding sizes for several values of $n$ on the Facebook link prediction task. Quadratic functions approximating $\log(auc\_score)$ were plotted to allow for a better understanding of the results. It can be seen that for larger numbers of permutations ($n > 100$) results improve with embedding size, while for small ones ($n = 1$) they decrease. The plot also shows that $n = 10$ gives fairly robust values for all tested embedding sizes.

A further analysis can be seen in Table 4, which shows that graphs with more edges per node tend to get better results with larger values of $n$, while graphs with a smaller branching factor have better results with only one permutation ($n = 1$). Other factors also enter into account when choosing $n$, like graph size. For example, link prediction on the DBLP graph had its best results for $n = 10$, although its branching size was only 3.31. Further experiments on this parameter can be seen in Appendices B.2 and C.1.

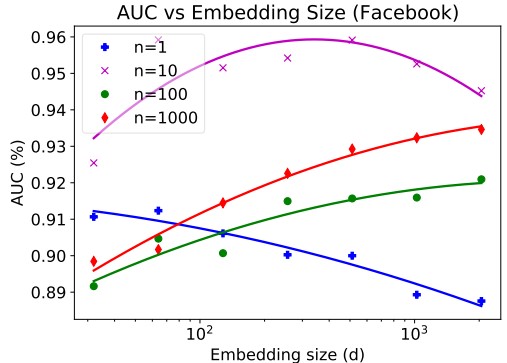

Figure 1: (Color online) NBNE AUC scores vs embedding sizes on Facebook dataset with 50% edges removes

Table 4: Link Prediction results for varying $n$ with NBNE

| $n$ | PPI (9.95[†]) | | | Facebook (21.84[†]) | | | Blog (32.38[†]) | | |
|---|---|---|---|---|---|---|---|---|---|
|  | Precision Train | Test | AUC | Precision Train | Test | AUC | Precision Train | Test | AUC |
| 10 | 0.7071 | 0.7108 | 0.7795 | **0.8453**[‡] | 0.9061 | 0.9642 | **0.8771**[‡] | 0.8627 | **0.9348**[‡] |
| 5 | 0.7280 | 0.7305 | 0.8071 | **0.8408**[‡] | **0.9070** | **0.9688** | **0.8775**[‡] | **0.8681** | **0.9375**[‡] |
| 1 | **0.7822** | **0.7751** | **0.8462** | 0.8036 | 0.8410 | 0.9150 | 0.8115 | 0.8374 | 0.9146 |

[†] Edges per node          [‡] No statistical difference

### 5.2 TIME COMPLEXITY

SkipGram's time complexity is linear on the number of sentences, embedding size ($d$) and logarithmic on the size of the vocabulary (Mikolov et al., 2013a). Since the number of sentences is linear on the number of permutations ($n$), branching factor of the graph ($b$) and on the number of nodes,

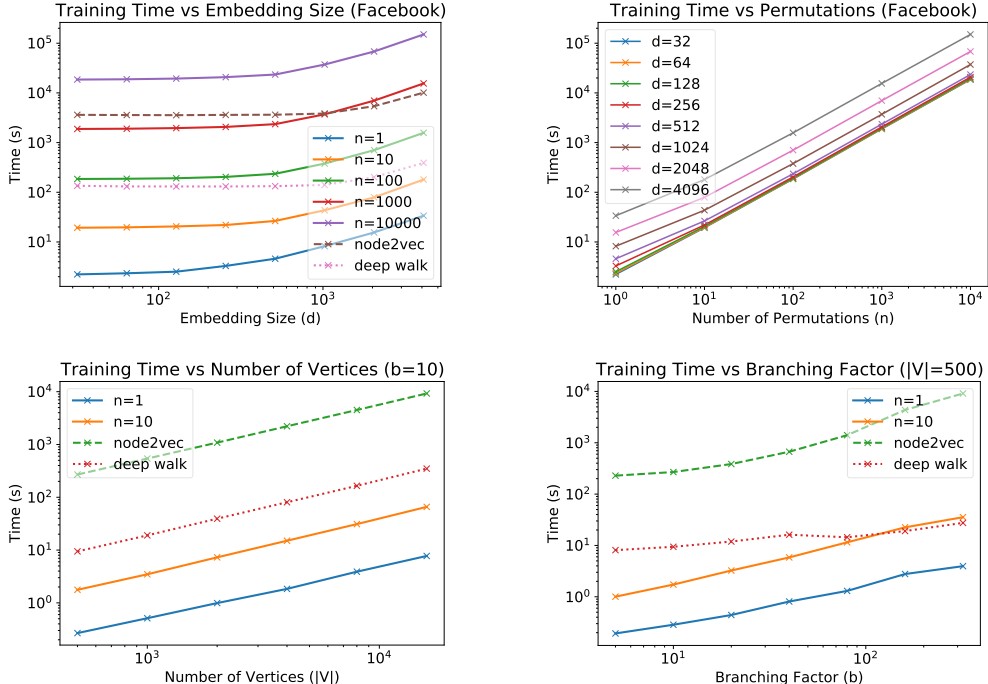

Figure 2: (Color online) **(Top)** Facebook dataset with 30% edges removed: Training times vs **(Left)** embedding size **(Right)** number of permutations. **(Bottom)** Randomly generated graphs: Training times vs **(Left)** number of vertices with $b = 10$ **(Right)** branching factor with $|V| = 500$.

which is the size of the vocabulary ($|V|$), the algorithm will take a time bounded by:

$$O\left(d \cdot n \cdot b \cdot |V| \cdot log(|V|)\right)$$

Figure 2 (Top-Left and Top-Right) show training time is indeed linear on both embedding size and number of permutations. It also shows that Node2Vec is considerably slower than DeepWalk, and only has a similar training time to running NBNE with at least $n = 1000$. NBNE with $n < 10$ was by far the fastest algorithm.

NBNE, Node2Vec and DeepWalk run in a time bounded by $O(|V| \log |V|)$, as can be seen in Figure 2 (Bottom-Left). Figure 2 (Bottom-Right) shows that NBNE's time complexity is linear in the branching factor $b$, while Node2Vec's is quadratic. DeepWalk's running time is constant in this parameter, but for a graph with a larger branching factor, a higher number of walks per node should be used to train this algorithm, which would make it indirectly dependent on this factor.

## 6 CONCLUSIONS

The proposed node embedding method NBNE shows results similar or better than the state-of-the-art algorithms Node2Vec and DeepWalk on several different datasets. It shows promising results in two application scenarios: link prediction and node classification, while being efficient and easy to compute for large graphs, differently from other node embedding algorithms, such as LINE (Tang et al., 2015) or SDNE (Wang et al., 2016).

NBNE focuses learning on node's immediate neighbors, creating more ego-centric representations, which we suspect makes them more stable and faster to learn. Empirical results show that, although it has a similar time complexity, NBNE can be trained in a fraction of the time taken by DeepWalk (10 to 190 times faster) or Node2Vec (200 to 6,000 times faster), giving fairly robust results. Since embeddings are learned using only a node's immediate neighbors, we suspect it should also be easier to implement more stable asynchronous distributed algorithms to train them, and we leave this as future work.

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

## A  DATASETS

We used a total of six graph datasets to evaluate NBNE, with DeepWalk and Node2Vec being used as baselines. Next we briefly describe these datasets:

1. Facebook (McAuley & Leskovec, 2012): A snapshot of a subgraph of Facebook, where nodes represent users and edges represent friendships.

2. Astro (Leskovec et al., 2007): A network that covers scientific collaborations between authors whose papers were submitted to the Astrophysics category in Arxiv.

3. Protein-Protein Interactions (PPI) (Breitkreutz et al., 2008): We use the same subgraph of the PPI network for Homo Sapiens as in (Grover & Leskovec, 2016). This subgraph contains nodes with labels from the hallmark gene sets (Liberzon et al., 2011) and represent biological states. Nodes represent proteins, and edges indicate biological interactions between pairs of proteins.

4. Wikipedia (Mahoney, 2011): A co-occurrence network of words appearing in the first million bytes of the Wikipedia dump. Labels represent Part-of-Speech (POS) tags.

5. Blog (Zafarani & Liu, 2009): A friendship network, where nodes are bloggers and edges are friendships between them. Each node in this dataset has one class which is referent to the blogger's group.

6. DBLP (Yang & Leskovec, 2012): A co-authorship network where two authors are connected if they published at least one paper together.

Figure 3 shows the distribution of classes in the three datasets used for node classification. While Wikipedia has a long tailed distribution, with one class being present in almost $50\%$ of its nodes, PPI's probabilities are well distributed along the $49$ different possible classes.

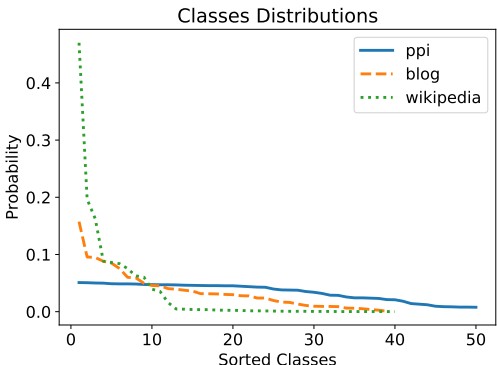

Figure 3: (Color online) Distribution of percentage of nodes per class in label classification datasets with classes sorted by their frequency.

## B  ASSORTATIVITY

Assortativity, also referred to as homophily in social network analysis, is a preference of nodes to attach themselves to others which are similar in some sense. In this section, we further investigate assortativity properties related to both the representations generated by our algorithm, as of the graphs themselves. In Section B.1, we do a quantitative analysis on the homophily inherent to the datasets considered in this work. In Section B.2, we make a qualitative analysis of how assortativity varies depending on the number of permutations $n$. In Section B.3, we make a qualitative analysis on the trade-off of first and second order proximities based on the choice of $k$.

## B.1 DATASETS' HOMOPHILY

There are several ways to quantitatively capture the homophily present in a graph. Jensen & Neville describe relational auto-correlation, which is Pearson's contingency coefficient on the characteristics of nodes which share edges (Jensen & Neville, 2002; Neville & Jensen, 2007). Park & Barabási (2007) define dyadicity and heterophilicity, which respectively measure how a graph's nodes share common/different characteristics in edges, compared to a random model.

Table 5 presents both degree and label assortativity properties of the six graphs analysed here, calculated using the definition of Newman (2003). We can see in this table that the datasets analyzed in this work cover a broad specter of assortativity properties. PPI, Wikipedia and Blog graphs present negative degree assortativity, which means nodes in these graphs are more likely to connect with nodes of different connectivity degrees. At the same time, Facebook, Astro and DBLP present positive degree assortativity, which indicates that their nodes tend to connect to others with similar degrees.

Table 5: Datasets homophily information

|  | Assortativity | |
|---|---|---|
|  | Degree[1] | Label[1] |
| Facebook | 0.0635 | - |
| Astro | 0.2051 | - |
| PPI | -0.0930 | 0.0533 |
| Wikipedia | -0.2372 | -0.0252 |
| Blog | -0.2541 | 0.0515 |
| DBLP | 0.2665 | - |

[1] Calculated as in (Newman, 2003)

We also analyze graphs with both positive and negative label assortativity in our label classification task. While PPI and Blog datasets present positive label assortativity, with connected nodes more frequently sharing classes, Wikipedia has a negative assortativity, with its connected nodes being more likely to have different classes.

## B.2 MORE ON THE NUMBER OF PERMUTATIONS ($n$)

Here, we further analyze how the number of permutations ($n$) influences both homophily and overfitting in our learned representations. We qualitatively measure homophily by comparing either cosine or euclidean distances between nodes on edges to the distances in non-edges.

The cosine distances for the PPI dataset, shown by the box plots in Figure 4 (top-left), clearly show for larger values of $n$ how the embeddings overfit to the specific graph structure, with the learned similarity on edges not generalizing to the links which were previously removed. In this graph, for larger numbers of permutation the removed edges have a distribution more similar to the non edges than to the edges used during training, which is a tendency that can be observed in the other graphs, although in a smaller scale.

The box plots in Figure 4 (top-right) show the cosine distance for Facebook nodes. We can see that for $n = 5$ there is a larger separation between removed edges and non edges, which justifies the algorithm's choice of this value. For larger values of $n$ we can again see an overlap between the distributions, caused by the embeddings overfitting. On the other hand, the cosine distances for the DBLP in Figure 4 (bottom-left) show the largest separation for $n = 10$.

Finally, the box plots in Figure 4 (bottom-right) show cosine distances for the Blog dataset. We can see that for $n = 1$ and $n = 5$ there is actually a larger cosine distance between nodes in removed edges than in non edges, with this situation only inverting for $n \geq 10$. This happens due to this graph's negative degree homophily. This is also observed for similar graphs in the PPI and Wikipedia datasets, though with a smaller intensity in the PPI graph, which has a smaller absolute value of degree assortativity and where only embeddings for $n = 1$ present this property.

The box plots from Figure 4 further support our intuition that graphs with larger branching factors should have larger values of $n$. At the same time, this choice also depends on the graph size and structure, as shown by the algorithms choice of $n = 10$ for the DBLP dataset, which contains the

largest degree assortativity. The best choice of $n$ depends on the analyzed task, but we believe that, at least for link prediction, this choice is both directly proportional to a graph's branching size and degree assortativity. Nonetheless, the difficulty in analyzing these graphs supports our choice for a semi-supervised approach, automatically choosing $n$ on a per graph instance.

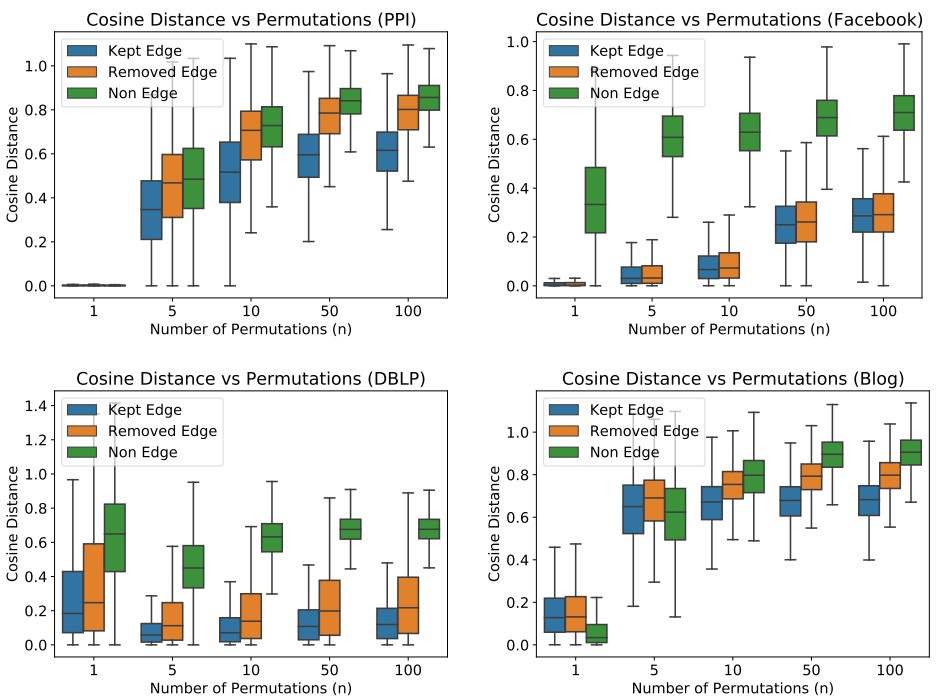

Figure 4: (Color online) Cosine distances on the: PPI dataset (top-left); Facebook dataset (top-right); DBLP dataset (bottom-left); Blog dataset (bottom-right). All graphs had 10% of edges removes.

Considering again the experiment on the PPI dataset with the number of permutations $n = 1$ in Figure 4 (top-left), in Figure 5 we present in detail the euclidean distances between nodes that share or not an edge for this number of permutations. We can see that the distribution of removed edges is a lot closer to the edges used for training than to the non edges.

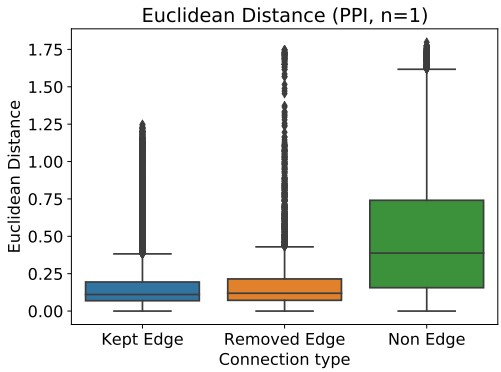

Figure 5: (Color online) Euclidean distances on PPI dataset for $n = 1$.

### B.3 TRADE-OFF BETWEEN FIRST AND SECOND ORDER PROXIMITY

The window size and the number of neighbors in a sentence are both adjusted by a single variable, $k$, and this variable also controls a trade-off between first and second order proximities in the node embeddings. This can be explained intuitively by analyzing both the sentence sampling method in Algorithm 1 and Equations 1, 2 and 3, in Section 3.2.

When a smaller $k$ is chosen, each node's embedding $r_{v_i}$ will be mainly used to predict its own neighbors. This causes nodes with shared neighbors to have closer representations (second order proximity). When larger values of $k$ are chosen, nodes will appear more often in its neighbors sentences, and will predict not only its own neighbors, but his neighbors' neighbors. This will result in connected nodes to have more similar embeddings, increasing first order similarity.

We further analyze this by examining the distribution of cosine distances between nodes at different graph distances. For this analysis, we use three different synthetic graphs: Barabási-Albert (Barabási & Albert, 1999); Erdõs-Rényi (Erdos & Rényi, 1960); Watts-Strogatz (Watts & Strogatz, 1998). We choose these graphs because of their structural differences, believing they cover an ample specter of different graphs' properties. These graphs were created with $|V| = 2000$ and $b = 20$, and Watts-Strogatz graphs had a probability $\beta = 0.2$ of generating non-lattice edges. To train our representations we used $n = 10$ and $d = 128$.

Figure 6 shows box plots of these cosine distances of nodes' representations versus their graph distance on these different artificial random graphs. In this figure, we can see that, for both Barabàsi-Albert and Erdõs-Rényi graphs, when using a sentence size ($k$) equal to 1, the cosine similarity is larger for nodes which are two steps away than for nodes which share an edge (second order proximity), while for larger values of $k$, nodes which share an edge have larger similarity (first order proximity).

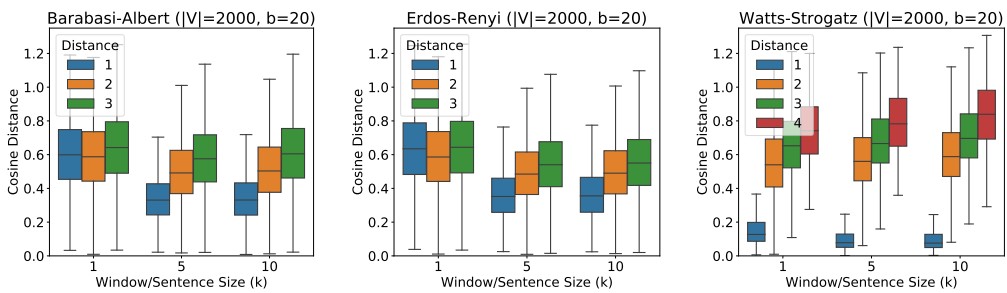

Figure 6: (Color online) NBNE features cosine similarities between nodes versus Graph Distance for different values of $k$ for the graphs Barabási-Albert (left), Erdõs-Rényi (middle) and Watts-Strogatz (right).

The box plots in Figure 6 also show that the difference in similarity increases with the value of $k$. The larger the value of $k$, the larger the difference between similarities of nodes which share an edge and nodes with larger distances, as can be seen in detail in Figure 7 for the Barabási-Albert graph.

## C GRAPH SIZE AND SPARSENESS ANALYSIS

In this section, we analyze how a graph's sparseness (represented here by its branching factor) and size (represented here by its number of vertices) affect the choice of the number of permutations ($n$) and of the window/sentence size ($k$). With this purpose we ran several link prediction experiments on two different synthetic graphs: Watts-Stogratz and Barabási-Albert.[6] These graphs were generated for different sizes ($|V|$) and sparseness ($b$), and we ran experiments with the same setup as in Section 4.1, with Watts-Stogratz graphs having again $\beta = 0.2$.[7] Section C.1 presents this analysis for the

---

[6] Erdõs-Rényi graphs weren't analyzed in this section because, since they have a completely random structure, its removed edges would be unpredictable.

[7] Results presented in this section are all averages of ten cross-validation executions in a single instance of each graph size.

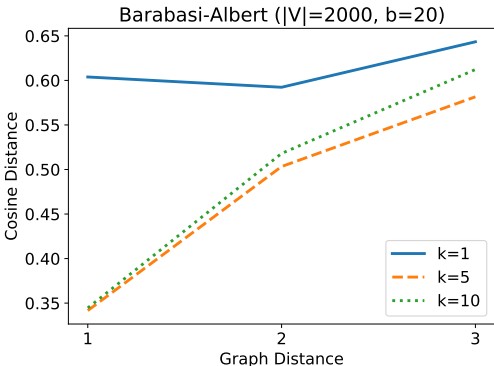

Figure 7: (Color online) NBNE cosine vs graph distances in the Barabási-Albert graph.

number of permutations ($n$) and Section C.2 contains the analysis for different window/sentence sizes ($k$).

## C.1   NUMBER OF PERMUTATIONS ($n$)

We analyze, in this section, how a graph's size and sparseness affect the choice of the number of permutations ($n$), for both Watts-Stogratz and Barabási-Albert graphs. Analyzing the graphs in Figure 8, we see a correlation between the best choice of $n$ and a graph's number of vertices ($|V|$) and branching factor ($b$). In Figure 8a, which contains the experiments in the most sparse graphs, results for $n = 1$ are better for all graph sizes. A random algorithm would return an AUC score of $0.5$, so results bellow this value clearly expose a problem in the learning algorithm. This is the case for both $n = 10$ and $n = 5$ in these graphs, which overfit its representations.

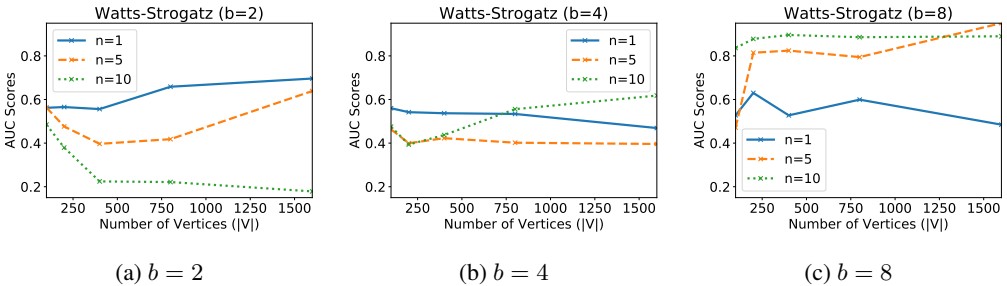

(a) $b = 2$        (b) $b = 4$        (c) $b = 8$

Figure 8: (Color online) AUC Score vs Number of Vertices on a link prediction task on synthetic Watts-Strogatz graphs for different values of $n$.

In Figure 8b we can see that, when considering a graph with a branching size of $b = 4$, for smaller graphs a smaller value of $n$ is preferable, while for larger graphs a larger number of permutations gives better results ($n = 10$). In Figure8c we can see that, for a branching size of $b = 8$, results for larger values of $n$ are always better than for $n = 1$. Notice also that, while results for $b = 2$ and $b = 4$ were around $0.55 \sim 0.7$, results for $b = 8$ are closer to $0.9$, showing that this algorithm is better at learning with more information.

Our experiments in link prediction using synthetic Barabási-Albert graphs present slightly more complex results. Figure 9 shows that for smaller branching factors ($b \leq 8$), $n = 1$ indeed generate better results for small graphs, but for larger graphs, a larger number of permutations is necessary. For intermediary branch sizes the best value of $n$ is harder to determine, and only for $b = 64$ we start to see a tendency of larger number of permutations consistently giving better results.

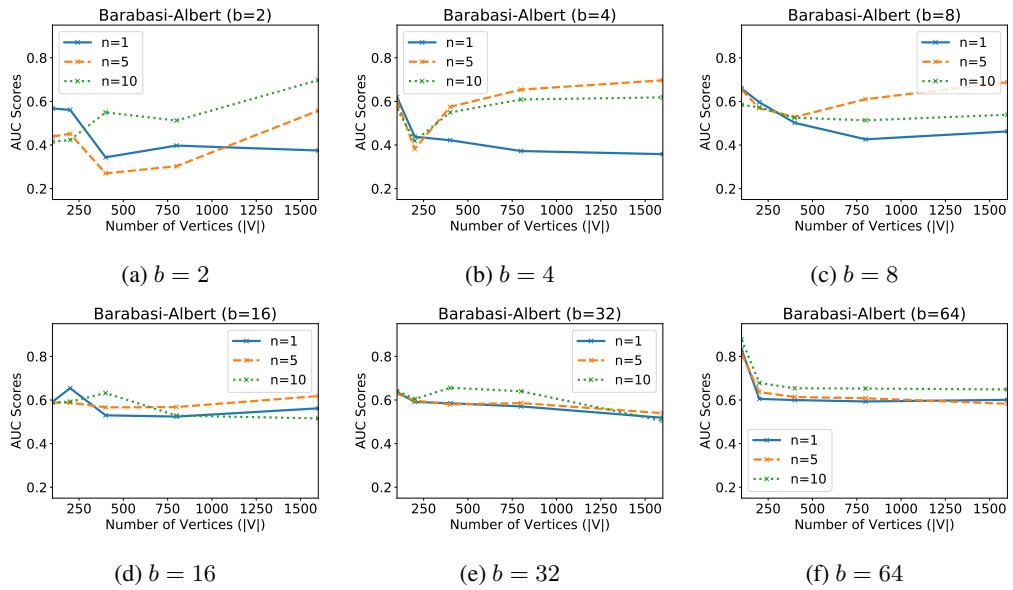

Figure 9: (Color online) AUC Score vs Number of Vertices on a link prediction task for synthetic Barabási-Albert graphs for different values of $n$.

We can also see from Figure 9 that edges in Barabási-Albert graphs are considerably more difficult to predict, specially for smaller branching sizes. Most of our results are around 60% and our best AUC scores in these graphs are around 70%.

Again, $n$'s dependency on these graph properties ($|V|$ and $b$) depends highly on the graph's structure, further supporting our choice of a semi-supervised approach, choosing $n$ on a per graph instance by validating results on a small validation set. This can be considered as a form of early stopping when training these node embeddings.

## C.2 Window and Sentence Sizes ($k$)

In this section, we again use Watts-Stogratz and Barabási-Albert graphs, this time to analyze how a graph's size and sparseness affect results for different window and sentence sizes ($k$) in our model. For these experiments we keep $n = 5$ fixed.

Figure 10a shows that, for a small branching factor ($b = 2$), all choices of $k$ clearly overfit for Watts-Strogatz graphs, but $k = 5$ overfits less than larger choices of $k$. For $b = 8$, $k = 5$ produces slightly better results in these graphs, while larger values of $k$ produce better results for a larger branching size ($b = 32$).

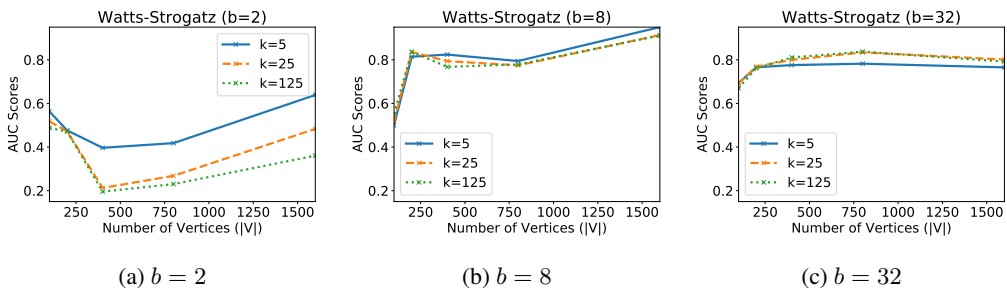

Figure 10: (Color online) AUC Score vs Number of Vertices on a link prediction task for synthetic Watts-Strogatz graphs for different values of $k$.

Barabási-Albert graphs' edges are considerably harder for our algorithm to predict, as shown in the previous section, so we only report results for larger values of $b$ (the algorithm, with our choice of hyper-parameters, overfits for smaller values). We can see from Figure 11 that larger values of $k$ usually produce better results for this graph, but are more propense to overfit, specially when being applied to larger sparse graphs ($|V| \geq 800$ and $b = 16$).

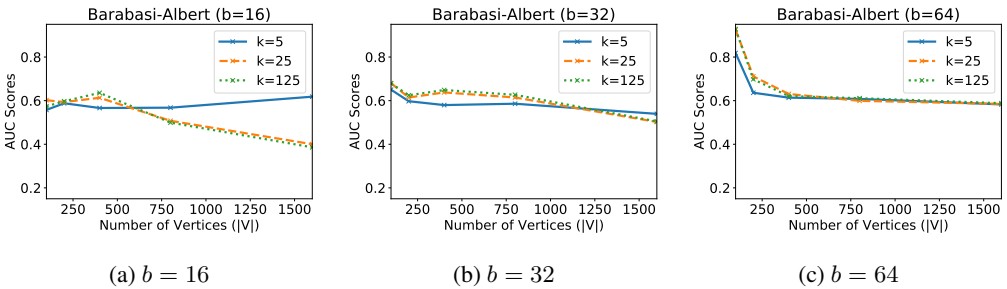

(a) $b = 16$        (b) $b = 32$        (c) $b = 64$

Figure 11: (Color online) AUC Score vs Number of Vertices on a link prediction task for synthetic Barabási-Albert graphs for different values of $k$.

Further analysis on the representations' properties for different values of $k$ could provide better motivation on its choice, but we leave this to future studies, keeping our choice of $k = 5$ constant in this work. Studying if geometric operations between representations have comprehensible meanings would also be interesting, such as was done for Word2Vec algorithms, but this is also left as future work.

## D    COMPARISON WITH SDNE

Structural Deep Network Embedding (Wang et al., 2016) is another algorithm used for learning node embeddings in graphs. As described in Section 2, SDNE is based on first and second order proximities, using autoencoders to learn compact representations based on a node's adjacency matrix (second-order proximity), while forcing representations of connected nodes to be similar (first-order proximity) by using an hybrid cost function.

This algorithm has a time complexity of $O(|V|^2)$, but its main computation, which is calculating the gradients of its cost function and updating model parameters, can be highly parallelized by using modern GPUs and Deep Learning frameworks. In this section, we compare NBNE and SDNE in terms of both efficiency and efficacy, analysing both AUC/Macro F1 scores and training time. With this objective, we trained SDNE embeddings using both a dedicated K40 GPU with CUDA 8.0 and a dedicated 16 core linux server.[8]

In their original work, SDNE was run in a semi-supervised setting, finding the best value of $\alpha$, $\beta$ and $\nu$ by tuning them on a small validation set. In this work we fix $\alpha = 0.2$ and $\beta = 10$, since in their work they state that these values commonly give the best results, while only choosing $\nu$ in a semi-supervised manner. We use SDNE's architecture with $[10,300; 1,000; 128]$ nodes on each layer and test it on both Link Prediction and Node Classification tasks, using the same steps described in Sections 4.1 and 4.2. We train these embeddings using $\nu \in \{0.1, 0.01, 0.001\}$ and choose the best value on the same validation sets used to tune $n$ for NBNE and $p$ and $q$ for Node2vec.

Table 6 shows results using both NBNE and SDNE embeddings on Link Prediction tasks. In this table we can see that both algorithms produce similar results in terms of AUC scores, with each having a statistically significant better result on two datasets, and NBNE having a non statistically significant, but slightly better result on the fifth. It is clear that even when training SDNE using a K40 GPU, NBNE still has more than an order of magnitude faster training time on all datasets, being more than two orders of magnitude faster on most. When comparing to SDNE trained on a CPU, NBNE has more than three orders of magnitude faster training time. On Astro, the dataset with the

---

[8] SDNE code was implemented using Tensorflow (Abadi et al., 2015)

largest number of nodes analyzed here, NBNE had a 2,009 times faster training time compared to SDNE on a GPU and 44,896 times faster compared to SDNE on CPU.[9]

Table 6: Link prediction results with SDNE

| | Facebook | | Astro | | PPI | |
|---|---|---|---|---|---|---|
| | AUC | Training Time | AUC | Training Time | AUC | Training Time |
| NBNE | **0.9688** | **0m11s‡** | **0.8328** | **0m07s‡** | 0.8462 | **0m02s‡** |
| SDNE | 0.9510* | 20m34s† 242m10s‡ | 0.8157* | 234m24s† 5,237m59s‡ | **0.8751*** | 16m10s† 232m01s‡ |
| Gain | 1.87% | 112.21x 1,320.91x | 2.10% | 2,009.17x 44,896.96x | -3.30% | 485.10x 6,960.34x |

| | Wikipedia | | Blog | | DBLP | |
|---|---|---|---|---|---|---|
| | AUC | Training Time | AUC | Training Time | AUC | Training Time |
| NBNE | **0.6853** | **0m02s‡** | 0.9375 | **1m11s‡** | **0.9335** | **14m30s‡** |
| SDNE | 0.6781 | 22m23s† 337m47s‡ | **0.9462*** | 81m33s† 1,492m47s‡ | - | - |
| Gain | 1.06% | 671.59x 10,133.46x | -0.92% | 68.92x 1,261.51x | - | - |

‡ Training time on CPU
† Training time on GPU

Table 7 shows the results of running NBNE and SDNE on the Node Classification task. On this task NBNE gave statistically better results on two datasets, with an impressive gain of 29.27% on PPI and 46.94% on Blog, only losing on Wikipedia with an also large gain of $-20.20\%$. We can again see that NBNE has a more than an order of magnitude faster training time than SDNE on a GPU in this dataset, being more than two orders of magnitude faster when SDNE is trained on a CPU.

Table 7: Node classifications results with SDNE

| | Blog | | PPI | | Wikipedia | |
|---|---|---|---|---|---|---|
| | Macro F1 | Training Time | Macro F1 | Training Time | Macro F1 | Training Time |
| NBNE | **0.2005** | **1m57s‡** | **0.0978** | **0m16s‡** | 0.0727 | **0m41s‡** |
| SDNE | 0.1364* | 96m48s† 1,476m33s‡ | 0.0757* | 16m52s† 231m04s‡ | **0.0911*** | 19m60s† 338m40s‡ |
| Gain | 46.94% | 49.64x 757.20x | 29.27% | 63.24x 866.48x | -20.20% | 29.26x 495.60x |

‡ Training time on CPU
† Training time on GPU

Analyzing both these tables we can also see that the largest gains in training time occur when using NBNE on a large but sparse network, such as Astro. This agrees with our theoretical expectations, since SDNE's time complexity grows quadratically with the number of nodes $O(|V^2|)$ and NBNE's grows with $O(|V| \cdot log(|V|) \cdot b)$, which is close to linear on the number of nodes for large graphs.

## E  FULL RESULTS

In this section, we extend the results presented in Section 4, considering now the precision on the training and test sets. We present results for the link prediction problem in Table 8 and for the node classification problem in Table 9.

NBNE produces statistically similar results to its baselines, in terms of Macro F1, on the node classification task using both PPI and Wikipedia datasets, while showing a statistically significant

---

[9]We tried running SDNE with the DBLP dataset, but after five days it hadn't reached half of the training, so we stopped it.

Table 8: Link prediction complete results

| **Facebook** | Precision | | AUC | Training Time |
|---|---|---|---|---|
| | Train | Test | | |
| NBNE | 0.8408 | 0.9070 | 0.9688 | **0m11s** |
| DeepWalk | 0.8770* | 0.9218 | 0.9730 | 2m26s |
| Node2vec | **0.8844***| **0.9251** | **0.9762** | 69m33s |
| Gain | -4.93% | -1.95% | -0.76% | (12.96x, 369.85x) |

| **Astro** | Precision | | AUC | Training Times |
|---|---|---|---|---|
| | Train | Test | | |
| NBNE | **0.7640** | **0.7552** | **0.8328** | **0m07s** |
| DeepWalk | 0.6957* | 0.6836* | 0.7548* | 6m55s |
| Node2vec | 0.7223* | 0.7163* | 0.7738* | 182m16s |
| Gain | 5.78% | 5.43% | 7.62% | (59.06x, 1555.80x) |

| **PPI** | Precision | | AUC | Training Times |
|---|---|---|---|---|
| | Train | Test | | |
| NBNE | **0.7822** | **0.7751** | **0.8462** | **0m02s** |
| DeepWalk | 0.7124* | 0.7078* | 0.7741* | 2m30s |
| Node2Vec | 0.7332* | 0.7253* | 0.7841* | 66m37s |
| Gain | 6.69% | 6.86% | 7.91% | (77.43x, 2061.67x) |

| **Wikipedia** | Precision | | AUC | Training Times |
|---|---|---|---|---|
| | Train | Test | | |
| NBNE | 0.6823 | **0.6223** | **0.6853** | **0m02s** |
| DeepWalk | 0.8024* | 0.5245* | 0.6534* | 7m38s |
| Node2Vec | **0.8129***| 0.5317* | 0.6547* | 236m60s |
| Gain | -16.07% | 17.04% | 4.67% | (194.86x, 6049.77x) |

| **Blog** | Precision | | AUC | Training Times |
|---|---|---|---|---|
| | Train | Test | | |
| NBNE | **0.8775** | **0.8681** | **0.9375** | **1m11s** |
| DeepWalk | 0.8560* | 0.8337* | 0.9098* | 28m13s |
| Node2Vec | 0.8664* | 0.8460* | 0.9202* | 838m41s |
| Gain | 1.28% | 2.61% | 1.88% | (23.86x, 709.24x) |

| **DBLP** | Precision | | AUC | Training Times |
|---|---|---|---|---|
| | Train | Test | | |
| NBNE | **0.9724**[†] | **0.8781**[†] | **0.9335**[†] | **14m30s** |
| DeepWalk | 0.9344[‡] | 0.8369[‡] | 0.9242[‡] | 164m34s |
| Node2Vec | 0.9449[‡] | 0.8498[‡] | 0.9322[‡] | 3,285m59s |
| Gain | 2.91% | 3.33% | 0.13% | (11.34x, 226.52x) |

† average of 10 fold results
‡ no statistical tests were run, due to the time necessary to run one fold

22.45% gain in the Blog dataset. Node classification results in the PPI dataset had the smallest precision among all datasets, with only approximately 14%, but there are 49 classes in it and a classifier which always guessed the most common class would only get 2.95% precision.

NBNE only shows a statistically worse result in test precision for node classification on the Wikipedia dataset, losing to DeepWalk, but having an order of magnitude faster training time than DeepWalk and more than two orders of magnitude faster training time than Node2Vec. On all other experiments in either node classification or link prediction it presented either statistically better or similar results to its baselines, while showing much faster training times.

Table 9: Node classification complete results

| Blog | Precision | | Macro F1 | Training |
|------|-----------|-----|----------|----------|
| | Train | Test | | Times |
| NBNE | **0.4235** | **0.3290** | **0.2004** | **1m57s** |
| DeepWalk | 0.3726* | 0.3108* | 0.1451* | 31m31s |
| Node2vec | 0.4022* | 0.3257 | 0.1637* | 959m12s |
| Gain | 5.28% | 1.00% | 22.45% | (16.18x, 492.57x) |

| PPI | Precision | | Macro F1 | Training |
|------|-----------|-----|----------|----------|
| | Train | Test | | Times |
| NBNE | 0.3930 | 0.1436 | 0.0978 | **0m16s** |
| DeepWalk | 0.4143* | **0.1457** | **0.0991** | 3m04s |
| Node2Vec | **0.4599*** | 0.1371 | 0.0971 | 83m02s |
| Gain | -14.54% | -1.45% | -1.35% | (11.82x, 319.78x) |

| Wikipedia | Precision | | Macro F1 | Training |
|-----------|-----------|-----|----------|----------|
| | Train | Test | | Times |
| NBNE | **0.5853** | 0.4938 | **0.0727** | **0m41s** |
| DeepWalk | 0.5595* | **0.5078*** | 0.0679 | 13m04s |
| Node2Vec | 0.5796* | 0.5002 | 0.0689 | 408m00s |
| Gain | 0.99% | -2.76% | 5.56% | (19.04x, 594.62x) |

# F  AUTHOR NAME DISAMBIGUATION

One of the hardest problems faced by current scholarly digital libraries is author name ambiguity (Ferreira et al., 2012). This problem occurs when an author publishes works under distinct names or distinct authors publish works under similar names (Ferreira et al., 2015). Automatic solutions, which are effective, efficient and practical in most situations, are still in need (Santana et al., 2014). In this section, we test our algorithm against the case where distinct authors publish works under similar names.

## F.1  EXPERIMENTAL SETUP

For this problem, the DBLP repository was crawled and the profiles of fourteen of the most prolific ambiguous authors were obtained, together with their direct co-authors' complete profiles. With this, we created a new dataset, called here DBLP-ambiguous, consisting of fourteen separate co-authorship networks (14 separate graphs), each with all the connections of one of these homonymous authors and their co-authors' connections. Details on these graphs can be seen in Table 10.[10]

Table 10: DBLP-ambiguous (DBLP-amb) Dataset Details

| Name | # Authors | Nodes | Edges |
|------|-----------|-------|-------|
| JingLi | 4 | 105,746 | 589,367 |
| JingWang | 16 | 108,913 | 581,457 |
| JunLiu | 2 | 106,533 | 590,032 |
| JunWang | 21 | 121,511 | 691,705 |
| JunZhang | 16 | 116,497 | 631,738 |
| LeiZhang | 38 | 118,798 | 664,898 |
| LiZhang | 14 | 122,403 | 693,916 |
| WeiLi | 57 | 157,427 | 887,727 |
| WeiWang | 85 | 183,962 | 1,103,702 |
| WeiZhang | 52 | 131,200 | 722,272 |
| XiaodongWang | 3 | 50,854 | 284,733 |
| XinWang | 14 | 107,920 | 578,084 |
| YangLiu | 33 | 130,319 | 740,501 |
| YuZhang | 9 | 131,683 | 734,214 |

---

[10]Link to used DBLP crawler and the dataset will be made available in final publication.

Using these co-authorship networks, embeddings were obtained by training on the graphs with 20% of the papers from each ambiguous author removed. After the embeddings had already been learned for each author, the probability of each possible author-coauthors "sentence" was calculated as:

$$s_{possible\_author} = [v_{possible\_author}, \, v_{coauthor\_1}, \, ..., \, v_{coauthor\_j}]$$

This probability is given by:

$$p(author) = \frac{1}{T} \sum_{t=1}^{T} \left( \sum_{-k \leq j \leq k, j \neq 0} (\log (p (v_{t+j}|v_t))) \right) \quad (4)$$

where $v_1 = author$, which comes from the NBNE model itself.

As a baseline, we used the typical solution that classifies the closest of the possible ambiguous authors as co-author for each of the test papers. If no path on the graph existed to any of the possible ambiguous authors, or if there was a tie between the distances to two or more of them, a random one was chosen between the possible ones. DeepWalk and Node2Vec were not used as baselines for this task due to the size of the 14 graphs analyzed here, most with more than 100,000 nodes and 500,000 edges, which would result in a prohibitive training time.

### F.2 EXPERIMENTAL RESULTS

Table 11 presents the results for the author name disambiguation task for each chosen author. This experiment was run using NBNE as an unsupervised algorithm with a fixed number of permutations $n = 10$, having no validation set. We also used sentences of size $k = 5$ and node embeddings of size $d = 128$.

Table 11: Author name disambiguation results

| Name | # Authors | Algorithm | Precision |
|---|---|---|---|
| Jing Li | 4 | NBNE | 0.9415 |
| | | Baseline | 0.9415 |
| JingWang | 16 | NBNE | 0.8791 |
| | | Baseline | 0.8512 |
| JunLiu | 2 | NBNE | 0.9709 |
| | | Baseline | 0.9651 |
| JunWang | 21 | NBNE | 0.8357 |
| | | Baseline | 0.7821 |
| JunZhang | 16 | NBNE | 0.8206 |
| | | Baseline | 0.8130 |
| LeiZhang | 38 | NBNE | 0.8843 |
| | | Baseline | 0.8309 |
| LiZhang | 14 | NBNE | 0.8661 |
| | | Baseline | 0.8201 |
| WeiLi | 57 | NBNE | 0.8221 |
| | | Baseline | 0.7822 |
| WeiWang | 85 | NBNE | 0.8143 |
| | | Baseline | 0.8070 |
| WeiZhang | 52 | NBNE | 0.8408 |
| | | Baseline | 0.8184 |
| XiaodongWang | 3 | NBNE | 0.9697 |
| | | Baseline | 0.9576 |
| XinWang | 14 | NBNE | 0.8639 |
| | | Baseline | 0.8639 |
| YangLiu | 33 | NBNE | 0.7955 |
| | | Baseline | 0.7540 |
| YuZhang | 9 | NBNE | 0.9268 |
| | | Baseline | 0.9024 |
| Average | | NBNE | 0.8737 |
| | | Baseline | 0.8492 |

After the embeddings had already been learned for each author, which can be done off-line, the NBNE algorithm was faster in assigning the authors than its baseline. This occurred because it

only required computing the probability of each possible author-coauthors "sentence" ($p(s)$), while the baseline had to dynamically get the distance between the papers' co-authors and the possible authors.

It can be seen in Table 11 that for all but two authors the precision was higher when using the NBNE embeddings instead of the graph baseline, while for the other two precision score remained the same.

