# OpenReview forum: "Fast Node Embeddings: Learning Ego-Centric Representations"
_ICLR.cc/2018/Conference — Invite to Workshop Track_

### Official Review · AnonReviewer1 · 2017-11-26
**The authors propose a node embedding model NBNE, which preserves both first and second order proximity. NBNE is similar to random-walk-based methods (DeepWalk, Node2Vec) while having a different definition of “walk”. It defines “neighbors of a node” to be the “walk”. It also proposes that the “sentence length” should be the same as “window size”. It outperforms baseline models (DeepWalk and Node2Vec) in link prediction and classification tasks and also significantly reduces training time.**

**Rating:** 5
**Confidence:** 4

**Review:**

This paper demonstrates good experiment results on several tasks. There are some pros and comes as below:

Pros
The proposed model provides a new view to generate training examples for random-walk-based embedding models.
Experiments are conducted on several datasets (6 datasets for link prediction and 3 datsets for classification).
Various experiments are provided to support the analysis of time complexity of the proposed model.
Additional experiments are provided in appendix, though the details of experimental setups are not provided.

Cons:
1. The novelty is limited. The main contribution is the idea of substituting random-walk by neighbors of a node. The rest of the model can be viewed as DeepWalk which requires walk length be the same as window size.
2. The experiment setup is not fair to the competitors. It seems that the proposed model is turned by validation, and the competitors adopt the parameters proposed in Node2Vec. A fair experiment shall require every model to turn their parameters by the validation dataset.
3. Furthermore, since embedding training is unsupervised, in reality no validation data can be used to select the parameters. Therefore a fair experiments it to find a universal set of parameters and use them across different datasets.

4. In section 4.1 (Link Prediction), is there negative sampling during training and testing? Otherwise the training and testing instances are all positive. Also, what is the ratio of positive instances over negative ones?
5. In section 5.1 (Number of Permutations), why is “test accuracy” adopted but not “AUC” (which is the evaluation metric in section 4.1)?
6. In section 5.1 (Number of Permutations), table 4 should contain only the results of the same task (either Link Prediction or Classification but not both).
7. Didn't compare with the state-of-the-art node embedding models.

---

> ### Author Response · Authors · 2017-12-20
> **Clarifications and new comparison with SDNE**
>
> Thank you for your review and suggestions.
> Our answers to your considerations are:
>
>
> 1. We believe the two algorithms have different motivations, with NBNE following a Breath First Search (BFS) strategy, while DeepWalk follows a strategy similar to Depth First Search (DFS). Although they may be similar, these strategies result in very different algorithms and produce embeddings with different properties.
>
> 2. Node2Vec in the original work by Grover and Leskovec is also semi-supervised and is also tuned on the validation set to choose the best values for both parameters p and q, so comparisons to this algorithm are already fair in this sense.
> We added a new paragraph on page 5 to make this description clearer: "On both these tasks [Link Prediction and Node Classification], DeepWalk and Node2Vec were used as baselines, having been trained and tested under the same conditions as NBNE and using the parameters as proposed in (Grover and Leskovec, 2016). More specifically, we trained them with the same training, validation and test sets as NBNE and used a window size of 10 (k), walk length (l) of 80 and 10 runs per node (r). For Node2Vec, which is a semi-supervised algorithm, we tuned p and q on the validation set, doing a grid search on values p,q in {0.25; 0.5; 1; 2; 4}."
>
> 3. Several state-of-the-art embedding training algorithms are actually semi-supervised. While Deep Walk and LINE are unsupervised, Node2Vec chooses p and q based on results in a validation set. SDNE also selects alpha, beta and nu by evaluating them on a validation set.
>
> 4. When training and testing the logistic regression we use both positive and negative samples of edges, having both groups with equal sizes (notice that during training parts of the removed edges could have been randomly included as negative samples). We used logistic regressions only to benchmark the power of the representations, not entering in the harder topic of working with unbalanced datasets (much more non edges than edges in a graph).
> We changed the text to make this clearer.
>
> 5. We hadn't noticed this and have changed the graphs to use AUC scores. Accuracy and AUC have similar trends, leading to the same conclusion in Section 5.1.
>
> 6. We also changed this table to only show Link Predictions results. Again, results lead to the same conclusion as before.
>
> 7. We added results for SDNE in Appendix D of our paper. We ran it for all datasets on both Link Prediction and Node Classification, except DBLP because of its size. We ran SDNE for alpha=0.2 and beta=10, since they seem to be the best parameters as per their analysis. We chose an architecture of size [10300-1000-100] for all our experiments, which is their architecture chosen for the BlogCatlog dataset, the only one we also use. Like in their paper, we run it as a semi-supervised algorithm, tuning nu in a validation set, choosing nu from {0.1, 0.01, 0.001}. They don't mention how they chose this value, so we selected it from {0.1, 0.01, 0.001} to test an ample set of values. In Link Prediction, both our algorithms perform similarly, with ours having better results in three datasets and theirs in two, but our algorithm has more than two orders of magnitude faster training when SDNE is on a GPU, and is three to four orders of magnitude faster when SDNE is trained in a CPU. On Node Classification we win in two of the three datasets, with a gain of 45% on blog. In this task NBNE is 29~63 times faster training than SDNE on a GPU and 495~866 times faster than SDNE on a CPU.
> If you have other suggestions of state-of-the-art algorithms to compare, we would be willing to run experiments.
>
>
> We would again like to thank you for your review. The clarifications of Section 4 and 4.1 make our experiments clearer to understand.
> The new comparison with state-of-the-art algorithm SDNE also makes it easier to compare both algorithms directly and shows that, although both algorithms are competitive, NBNE usually produces better results in terms of AUC/Macro F1 scores. At the same time it shows that NBNE is much more computationally efficient, running in a fraction of the time taken by SDNE on both GPU or CPU.
>
> Furthermore, we also added two new sections in Appendix Sections B and C, per request of AnonReviewer2, with an in depth analysis of our algorithm considering: (i) an homophily analysis of both the datasets themselves and the learned representations and (ii) a series of tests analyzing results for different values of n and k on two synthetic graphs with various sizes (|V|) and connectedness (b). We believe these experiments give further support to our choice of a semi-supervised approach to choose n and  give a more solid understanding of how parameters n and k affect our resulting representations.
>
> If you have any other suggestions/questions, we would be pleased to answer them before the deadline of January 5th.

---

### Official Review · AnonReviewer3 · 2017-11-26
**Good work**

**Rating:** 6
**Confidence:** 4

**Review:**

The authors propose a method for learning node representations which, like previous work (e.g. node2vec, DeepWalk), is based on the skip-gram model. However, unlike previous work, they use the concept of shared neighborhood to define context rather than applying random walks on the graph.

The paper is well-written and it is quite easy to follow along with the discussion. This work is most similar, in my opinion, to node2vec. In particular, when node2vec has its restart probability set pretty high, the random walks tend to stay within the local neighborhood (near the starting node). The main difference is in the sentence construction strategy. Whereas node2vec may sample walks that have context windows containing the same node, the proposed method does not as it uses a random permutation of a node's neighbors. This is the main difference between the proposed method and node2vec/DeepWalk.

Pros:

Outperforms node2vec and DeepWalk on 5 of the 6 tested datasets and achieves comparable results on the last one.
Proposes a simple yet effective way to sample walks from large graphs.


Cons:

The description on the experimental setup seems to lack some important details. See more detailed comments in the paragraph below. While LINE or SDNE, which the authors cite, may not run on some of the larger datasets they can be tested on the smaller datasets. It would be helpful if the authors tested against these methods as well.

 For instance, on page 5 footnote 4 the authors state that DeepWalk and node2vec are tested under similar conditions but do not elaborate. In NBDE, when k=5 a node u's neighbors are randomly permuted and these are divided into subsets of five and concatenated with u to form sentences. Random walks in node2vec and DeepWalk can be longer, instead they use a sliding context window. For instance a sentence of length 10 with context window 5 gives 6 contexts. Do the authors account for this to ensure that skip-gram for all compared methods are tested using the same amount of information. Also, what exactly does the speedup in time mean. The discussion on this needs to be expounded.

---

> ### Author Response · Authors · 2017-12-20
> **Clarifications and new comparison with SDNE**
>
> Thank you for your review and suggestions.
> Our answers to your considerations are:
>
> "The description on the experimental setup seems to lack some important details. See more detailed comments in the paragraph below. While LINE or SDNE, which the authors cite, may not run on some of the larger datasets they can be tested on the smaller datasets. It would be helpful if the authors tested against these methods as well."
>
> --> We added results for SDNE in Appendix D of our paper. We ran it for all datasets on both Link Prediction and Node Classification, except DBLP because of its size. We ran SDNE for alpha=0.2 and beta=10, since they seem to be the best parameters as per their analysis. We chose an architecture of size [10300-1000-100] for all our experiments, which is their architecture chosen for the BlogCatlog dataset, the only one we also use. Like in their paper, we run it as a semi-supervised algorithm, tuning nu in a validation set, choosing nu from {0.1, 0.01, 0.001}. They don't mention how they chose this value, so we selected it from {0.1, 0.01, 0.001} to test an ample set of values. In Link Prediction, both our algorithms perform similarly, with ours having better results in three datasets and theirs in two, but our algorithm usually has more than two orders of magnitude faster training when SDNE is on a GPU, and is three to four orders of magnitude faster when SDNE is trained in a CPU. On Node Classification we win in two of the three datasets, with a gain of 46% on blog. In this task NBNE has a 29~63 times faster training than SDNE on a GPU and 495~866 times faster than SDNE on a CPU.
>
> "For instance, on page 5 footnote 4 the authors state that DeepWalk and node2vec are tested under similar conditions but do not elaborate."
>
> -->We added a new paragraph on page 5 to make this description clearer: "On both these tasks [Link Prediction and Node Classification], DeepWalk and Node2Vec were used as baselines, having been trained and tested under the same conditions as NBNE and using the parameters as proposed in (Grover and Leskovec, 2016). More specifically, we trained them with the same training, validation and test sets as NBNE and used a window size of 10 (k), walk length (l) of 80 and 10 runs per node (r). For Node2Vec, which is a semi-supervised algorithm, we tuned p and q on the validation set, doing a grid search on values p,q in {0.25; 0.5; 1; 2; 4}."
>
> "In NBNE, when k=5 a node u's neighbors are randomly permuted and these are divided into subsets of five and concatenated with u to form sentences. Random walks in node2vec and DeepWalk can be longer, instead they use a sliding context window. For instance a sentence of length 10 with context window 5 gives 6 contexts. Do the authors account for this to ensure that skip-gram for all compared methods are tested using the same amount of information. Also, what exactly does the speedup in time mean. The discussion on this needs to be expounded."
>
> --> We did not account for these changes, nor did we add an extra section with these experiments to the paper, because there are three different parameters in DeepWalk and Node2Vec which control the amount of information used: the number of runs per node (r); length of walks (l) and; window size (k). To find a value for these three variables together which at the same time compared in amount of computation to NBNE and gave good results would require a deep analysis on them, since they interact with one another in non-linear ways. At the same time, to run a semi-supervised version of both DeepWalk and Node2Vec which chose these parameters by evaluating them in a validation set would be too computationally expensive, specially since Node2vec already takes more than 800 minutes to train for a single of these values on Blog.
>
> We would again like to thank you for your review. The new comparison with SDNE makes it easier to compare both algorithms directly and shows that, although both algorithms are competitive, NBNE usually produces better results in terms of AUC/Macro F1 scores. At the same time, it shows that NBNE is much more computationally efficient, running in a fraction of the time taken by SDNE on both GPU or CPU.
>
> Furthermore, we also added two new sections in Appendix Sections B and C, per request of AnonReviewer2, with an in depth analysis of our algorithm considering: (i) an homophily analysis of both the datasets themselves and the learned representations and (ii) a series of tests analyzing results for different values of n and k on two synthetic graphs with various sizes (|V|) and connectedness (b). We believe these experiments give further support to our choice of a semi-supervised approach to choose n and  give a more solid understanding of how parameters n and k affect our resulting representations.
>
> If you have any other suggestions/questions, we would be pleased to answer them before the deadline of January 5th.

---

### Official Review · AnonReviewer2 · 2017-11-28
**The papers presents approach to embed nodes based on their 1-hop (i.e, immediate) neighbors.  It is not clear on what types of graphs the approach will perform well.  The space of graphs is not exhausted by the six chosen data sets.**

**Rating:** 4
**Confidence:** 5

**Review:**

The paper includes the terms first-order proximity ("the concept that connected nodes in a graph should have similar properties") and second-order proximity ("the concept that nodes with similar neighborhoods should have common characteristics"). These are called homophily in social network analysis. It is also known as assortativity in network science literature. The paper states on Page 4: "A trade-off between first and second order proximity can be achieved by changing the parameter k, which simultaneously controls both the sizes of sentences generated and the size of the wind used in the SkipGram algorithm." It is not readily clear why this statement should hold. Also the paper does not include a discussion on how the amount of homophily in the graph affects the results. There are various ways of measuring the level of homophily in a graph. There is simple local consistency, which is % of edges connecting nodes that have the same characteristics at each endpoint. Neville & Jensen's JMLR 2007 paper describes relational auto-correlation, which is Pearson contingency coefficient on the characteristics of endpoints of edges. Park & Barabasi's PNAS 2007 paper describes dyadicity and heterophilicity, which measures connections of nodes with the same characteristics compared to a random model and the connections of nodes with different characteristics compared to a random model.

k ("which simultaneously controls both the sizes of sentences generated and the size of the wind used in the SkipGram algorithm") is a free-parameter in the proposed algorithm. The paper needs an in-depth discussion of the role of k in the results. Currently, no discussion is provided on k except that it was set to 5 for the experiments.  From a network science perspective, it makes sense to have k vary per node.

It is also not clear why d = 128 was chosen as the size of the embedding.

From the description of the experimental setup for link prediction, it is not clear if a stratified sample of the entries of the adjacency matrix (i.e., both 0 and 1 entries) where selected.

For the node classification experiments, information on class distribution and homophily levels would be helpful.

In Section 5.1, the paper states: "For highly connected graphs, larger numbers of permutations should be chosen (n in [10, 1000]) to better represent distributions, while for sparser graphs, smaller values can be used (n in [1, 10])." How high is highly connected graphs? How spare is a sparser graph? In general, the paper lacks an in-depth analysis of when the approach works and when it does not.  I recommend running experiments on synthetic graphs (such as Barabasi-Albert, Watts-Strogatz, Forest Fire, Kronecker, and/or BTER graphs), systematically changing various characteristics of the graph, and reporting the results.

The faster runtime is interesting but not surprising given the ego-centric nature of the approach.

---

> ### Author Response · Authors · 2017-12-15
> **Clarifications and added two new in depth analysis on (i) homophily and (ii) results on synthetic graphs.**
>
> Thank you for the detailed review. Bellow, we address each point in your review.
> Furthermore, we added two new sections in the Appendix with an in depth analysis of our algorithm considering: (i) an homophily analysis of both the datasets themselves and the learned representations and (ii) a series of tests analyzing results for different values of n and k on two synthetic graphs with various sizes (|V|) and connectedness (b).
>
> Our answers to each of your considerations are:
>
> 1. First and second order proximity and k
>
> --> We added a new section in Appendix B.3 which includes an intuitive and a qualitative analysis of why this property holds. We do this analysis using three synthetic graphs: Barabasi-Albert, Erdos-Renyi and Watts-Strogatz.
>
> 2. Homophily in the results
>
> --> In Section B.1 of the Appendix we added a quantitative analysis of the homophily inherent to the datasets themselves. This analysis shows that we work with a diverse set of graphs, with both positive and negative degree and label assortativity. We also added an in depth qualitative analysis of homophily and overfitting of our learned representations in relation to n. These particular experiments led to very interesting results. In the plotted graphs we can clearly see the overfitting in our representations as the number n of permutations grows. We believe these experiments also support our choice for a semi-supervised approach to choose n.
>
> 3. k in the results
>
> --> We added a new section in Appendix C.2, in which we analyze results for different values of k. We used two different synthetic graphs in this analysis: Barabasi-Albert and Watts-Strogatz, creating them with different sizes and sparseness. We concluded that, although larger choices for this parameter (k={25,125}) give better results in several graphs, they are also more prone to overfitting, with k=5 being more robust.
>
> 4. d = 128
>
> --> We chose this value because it was used by our baselines in their works, to make our comparisons fair.
>
> 5. Link prediction: Stratified sample
>
> --> When training and testing the logistic regression we use both positive and negative samples of edges, having both groups with equal sizes (notice that during training parts of the removed edges could have been randomly included as negative samples). We used logistic regressions only to benchmark the power of the representations, not entering in the harder topic of working with unbalanced datasets (much more non edges than edges in a graph).
> We changed the text to make it clearer.
>
> 6. Node classification: Class distribution and homophily
>
> --> We added both degree and label homophily levels in Section B.1 in the Appendix, having in our tested datasets graphs with both positive and negative correlations. We also added the distribution of classes in each dataset in Appendix A. This plotted graph also shows the diversity in our analyzed datasets, containing both long tailed class distributions, in the Wikipedia dataset, and a more balanced distribution in the PPI dataset.
>
> 7. How connected/sparse?
>
> --> It is hard to state what is a highly connected or sparse graph in this sense, since these embeddings also depend on degree homophily levels, graph size and structure. But we added new experiments in Section C.1 on synthetic graphs to give more insight to answer this question. Our results show that indeed denser graphs tend to have better results with larger values of n, but how dense they should be depends on the graph structure. We believe this difficulty in selecting n without a deeper analysis of the graph justifies our choice for a semi-supervised algorithm, which can select the best value depending on results in a small validation set. This has the further advantage that n can be trained interactively with increasing values, not needing to be retrained for each new case, which is similar to early stopping in other machine learning settings.
>
> 8. Experiments on synthetic graphs
>
> --> As stated above, we addressed this problem doing an in depth analysis of both parameters n and k on Barabasi-Albert and Watts-Strogatz graphs, and it is present in Appendix C.
>
> 9. Faster runtime is interesting but not surprising
>
> --> We agree that the faster runtime is intuitive. But this faster runtime allied to the better/similar results in all tested datasets/experiments is a good contribution of our proposed method.
>
> We would like to thank you again for your detailed review and suggestions, specially concerning the suggested analysis on synthetic graphs and on homophily properties. They give our motivation for a semi-supervised algorithm a stronger experimental justification, and we added it as one of our contributions in the Introduction.
> We believe they were indeed important, giving a more solid understanding of our algorithm and increasing the paper's contributions.
>
> If you have any other suggestions/questions, we would be pleased to answer them and maybe try to conduct more experiments before the deadline of January 5th.

---

> > ### Author Response · Authors · 2017-12-20
> > **New comparison with SDNE**
> >
> > We added a comparison with SDNE in Appendix D of our paper, which was suggested by AnonReviewer1 and AnonReviewer3. We ran it for all datasets on both Link Prediction and Node Classification tasks, except DBLP because of its size, running SDNE with alpha=0.2 and beta=10, since they seem to be the best parameters as per their analysis. We chose an architecture of size [10300-1000-100] for all our experiments, which is their architecture chosen for the BlogCatlog dataset, the only one we also use. Like in their paper, we ran it as a semi-supervised algorithm, tuning nu in a validation set, choosing nu from {0.1, 0.01, 0.001}. They don't mention how they chose this value, so we selected it from {0.1, 0.01, 0.001} to test an ample set of values. In Link Prediction, both our algorithms perform similarly, with ours having better results in three datasets and theirs in two, but our algorithm usually has more than two orders of magnitude faster training when SDNE is on a GPU, and is three to four orders of magnitude faster when SDNE is trained in a CPU. On Node Classification we win in two of the three datasets, with a gain of 46% on blog. In this task NBNE has a 29~63 times faster training than SDNE on a GPU and 495~866 times faster than SDNE on a CPU.

---

### Decision · Program_Chairs · 2018-01-29
**ICLR 2018 Conference Acceptance Decision**

**Decision:**

Invite to Workshop Track

**Comment:**

The authors addressed the reviewers concerns but the scores remain somewhat low.
The method is not super novel, but it is an incremental improvement over existing approaches.